# Preparation of Alkaline Polyelectrolyte Membrane Based on Quaternary Ammonium Salt–Modified Cellulose and Its Application in Zn–Air Flexible Battery

**DOI:** 10.3390/polym13010009

**Published:** 2020-12-22

**Authors:** Xiang You, Congde Qiao, Dan Peng, Weiliang Liu, Cong Li, Hui Zhao, Hao Qi, Xiaoxia Cai, Yanqiu Shao, Xinhua Shi

**Affiliations:** 1School of Materials Science & Engineering, Qilu University of Technology (Shandong Academy of Sciences), Jinan 250353, China; y296405224@163.com (X.Y.); cdqiao@qlu.edu.cn (C.Q.); liuwl@qlu.edu.cn (W.L.); huizhaochem@163.com (H.Z.); qihao9709@163.com (H.Q.); 2Shandong Provincial Key Laboratory of Processing &Testing Technology of Glass and Functional Ceramics, Jinan 250353, China; 3Advanced Materials Institute, Qilu University of Technology (Shandong Academy of Sciences), Jinan 250014, China; lonarpeng@aliyun.com (D.P.); xhshi@qlu.edu.cn (X.S.); 4State Key Laboratory of Biobased Material and Green Papermaking, Qilu University of Technology, Shandong Academy of Sciences, Jinan 250353, China; congli2016@163.com; 5School of Chemical Engineering, Sichuan University, Chengdu 610065, China

**Keywords:** microcrystalline cellulose, polyvinyl alcohol, quaternary ammonium salt, alkaline polyelectrolyte membrane, Zn–air battery

## Abstract

In this study, a type of alkaline solid polyelectrolyte (ASPE) membrane was developed via the introduction of microcrystalline cellulose (MCC) and its modified product (QMCC) into the polyvinyl alcohol (PVA) matrix. In this process, green NaOH/urea-based solvent was used to achieve a good dispersion of MCC in the PVA matrix; meanwhile, the OH^−^ groups in the NaOH/urea-based solvent provided an alkaline environment for good ion conductivity. Compared to the MCC-incorporated ASPE, further improved conductivity was achieved when the MCC was modified with quantitative quaternary ammonium salt. TGA showed that the addition of QMCC improved the water retention of the matrix, which was beneficial to the OH^−^ conduction in the system. Compared to the control (50 mS cm^−1^), a maximum conductivity of 238 mS cm^−1^ was obtained after the incorporation of QMCC in the PVA matrix. Moreover, the tensile strength of the polymer electrolyte were also significantly increased with the addition of QMCC. Finally, this developed ASPE membrane was used in assembling a flexible Zn–air battery and showed a promising potential in the development of flexible electronic devices.

## 1. Introduction

With the rapid development of wearable electronic devices, there is an increasing demand for flexible batteries via various structure-based design strategies [1,2,3,4]. Flexible battery is a kind of new battery that can be deformed under certain external forces without impairing its performance [5,6,7,8]. Compared to conventional batteries, the key part of a flexible battery, i.e., the electrolyte, needs to meet higher requirements: such as smaller size, lighter mass, and bending resistance. In this context, a solid polymer electrolyte (SPE)—a kind of new electrolyte was developed [5]. Among these electrolytes, alkaline solid polyelectrolyte (ASPE) membrane is a promising substitute for the alkaline electrolyte in the application of alkaline batteries, in which ASPE plays a key role as an anion exchange membrane for the OH^−^ transportation. As one of the core components in battery, the alkaline polyelectrolyte is composed of polymeric materials and alkali metal salts [9,10]. The alkaline solid polyelectrolyte (ASPE) membrane plays roles in OH^−^ conduction and the separation of cathode and anode, endowing batteries with high conductivity, light weight, low cost, and environmental protection properties. In recent years, ASPE has been extensively studied, especially in the fields of super capacitor, lithium-ion battery, and sensor [8,11,12,13,14,15,16,17].

Generally, alkaline polyelectrolytes can be classified as aliphatic and aromatic types. The aliphatic type includes chitosan (CS)- [18,19], polyoxyethylene (PEO)- [20,21], and polyvinyl alcohol (PVA) [22,23]-based polyelectrolyte. Chitosan-based alkaline polyelectrolytes are readily available and biodegradable. However, due to low ion mobility and poor solubility, the application value of chitosan has been seriously restricted [19]. For PEO-based polymeric electrolytes, the large number of ether bonds in its main chain could interact with metal ions, which endows the polymeric electrolyte with better conductivity. However, the intrinsic crystallization within the PEO matrix hinders the effective transportation of OH^−^ in the polymer matrix and limits further improvement in conductivity [24]. In comparison, PVA-based alkaline polyelectrolytes contain a large amount of –OH groups with good hydrophilicity and low crystallinity, providing a possibility for further improving the conductivity. On the other hand, it is noteworthy that the poor alkali resistance of PVA makes the chemical structure prone to being destroyed in alkali environment. To improve the alkali resistance of PVA, some materials with alkali resistance properties can be applied via blending with the PVA. A series of substances with good alkali resistance, such as cellulose, have been used to enhancethe alkali resistance of PVA-based polyelectrolyte [10]. In terms of aromatic polymer electrolytes, such as polyetherimide(PEI) [25], polyarylethersulfone(PAES) [26], polyaryletherketone(PEK) [27], polyphenylene oxide (PPO) [28], polybenzimidazole (PBI) [29], etc., they are good at chemical stability and thermal stability; however, their intrinsic poor solubility reduces their value in polyelectrolyte application [30]. Hydroxides such as potassium-, lithium-, and sodium-based alkali metal salts are commonly used in alkaline polyelectrolytes. With the discovery that the ion migration number of KOH is much larger than that of NaOH and LiOH, KOH has become the first choice for researchers in the study of alkaline electrolyte membranes [31]. There are usually two ways to introduce KOH into an electrolyte membrane: one is to immerse the prepared polyelectrolyte membrane in a certain concentration of KOH solution [32], the other is to add KOH into the polymer compound directly [10]. No matter which introduction method is used, good alkali stability is necessary for the system, whereby a higher content of OH^−^ can be incorporated in the system to achieve improved conductivity [33,34,35,36,37,38].

In order to improve the alkali’s stability without compromising the OH^−^ conductivity of the polymeric matrix, a strategy was proposed in this paper to design the alkaline polyelectrolyte. Here, green NaOH/urea-based solvent was used for the dissolution of microcrystalline cellulose, and a quaternary ammonium salt (QAS)–grafted cellulose was prepared therefrom. A novel polyelectrolyte membrane was finally obtained via casting the blend of QAS-grafted cellulose with PVA in KOH solution [39]. As the most abundant natural resource, cellulose was selected herein due to the large number of hydroxyl groups grafted on the cellulose chain, which endows the material with good water retention capacity and alkaline resistance. This property is beneficial for the increase of OH^−^ and water contents, as well as the resulting conductivity [40,41]. After being modified by QAS, a further improved conductivity can be obtained due to the higher content of OH^−^ groups captured via QAS. Furthermore, the poor mechanical properties of PVA can be improved via cellulose under strong alkaline conditions owing to the intrinsic high strength of cellulose [16].

## 2. Materials and Methods

### 2.1. Materials

Polyvinyl alcohol (Mn = 80,000, AR, Chemical Reagent Factory of Tianjin, Tianjin, China), Microcrystalline cellulose (MCC) with a diameter of 20 µm (Sigma-Aldrich Co., St. Louis, MO, USA), epichlorohydrin (AR, Shanghai Macklin Biochemical Co., Ltd., Shanghai, China), dodecyl dimethyl tertiary amine (AR, Shanghai Macklin Biochemical Co., Ltd., China), polyvinylidene fluoride and N-methyl-2-pyrrolidone(Tianjin Baishi chemical Co. Ltd., Tianjin, China), potassium permanganate (AR, Tianjin Damao Chemical Co., Ltd, Tianjin, China).

### 2.2. General Characterization

Scanning electron microscopy (SEM, JSM-7500F, JEOL Ltd., Akishima-shi, Japan) was applied to observe the fracture surface morphology of the alkaline polymer electrolyte membrane. The specimens were freeze-dried, frozen under liquid nitrogen, then fractured, mounted, coated with gold, and observed. The infrared (FTIR) spectrum was measured in a Fourier infrared spectrometer (Nicolet 10, Thermo Fisher Scientific, Waltham, MA, USA) by KBr tableting. A TGA/SDTA851 system (Setaram, Caluire-et-Cuire, France) was used to analyze the polymer by thermogravimetry (TGA) under a nitrogen (N_2_) atmosphere at a flow rate of 10 °C/min. For PVA/KOH/QMCCx alkaline polymer electrolytes, the ionic conductivity was conducted using electrochemical impedance analyzer (CHI660E, Shanghai Chenhua Instruments Co., Shanghai, China) with an AC impedance method at room temperature. The AC frequency was scanned from 10^5^ to 10^−2^ Hz with an amplitude of 5 mV. Samples were sandwiched between two stainless steels (SS|SPE|SS) with a surface area of 0.785 cm^2^. The bulk resistance, R_b_, can be obtained via the cross point of the curves at the real axis. In terms of the ionic conductivity, calculation formula σ = L/(R_b_ × A) is used, where R_b_, A, and L represent bulk resistance (ohm), area (cm^2^), and the thickness (cm) of the sample, respectively. The electrochemical stability window was tested via a cyclic voltammetry curve using a CHI660E Electrochemical Workstation. The samples with a radius of 0.5 cm were sandwiched between two stainless steels (SS|SPE|SS). PVA/KOH/QMCCx alkaline polymer electrolyte membranes were tested in a voltage ranging from −0.5 V to 0.5 V and a scan rate of 10 mV s^−1^ at 25 °C. The charge–discharge performance of the battery was studied using a LAND auto-cycler (CT2001A, Wuhan Blue Electrical Co., Wuhan, China). The mechanical performance of the PVA/KOH/QMCCx alkaline polymer electrolyte membranes was tested using a mechanical testing machine (WDL-005, Jinan Xinshijin Experimental Instrument Co., Jinan, China) with a crosshead speed of 20 mm/min. The specimens with 30 mm length and 11 mm width were applied and coated with silicone wax to avoid the evaporation of water in the test.

### 2.3. Preparation of PVA/KOH Alkaline Polyelectrolyte Membrane

Briefly, 2.15g PVA was dissolved in 20 mL deionized water via mechanical stirring at 85 °C for 1 h, and 1 g glycerol was added into the solution dropwise with continuous stirring for 1 h until complete dissolution. Upon cooling to room temperature, 10 mL 9 M KOH solution was added dropwise with continuous stirring for 1 h. After removing bubbles via vacuum, the prepared solution was poured on a polytetrafluoroethylene mold and dried in the fume hood for hours, and a PVA/KOH polymeric electrolyte membrane was obtained. This prepared PVA/KOH membrane was rinsed with ethyl alcohol and deionized water several times and placed in a 50% humidity environment for water balance and later use.

### 2.4. Preparation of PVA/KOH/MCC Alkaline Polyelectrolyte Membrane

To obtain the cellulose solution, 7 wt.% NaOH and 12 wt.% urea (urea) were dissolved in deionized water. At room temperature, 1, 4, and 7 g MCC were added into NaOH/urea solution separately, which were uniformly dispersed by ultrasound and stirred at −12.3 °C until completely dissolved, thus obtaining a colorless, transparent cellulose solution.

Next, 2.15 g PVA was dissolved in 20 mL deionized water, mechanically stirring at 85 °C for 1 h, then 1 g glycerol was added into the solution, stirring for 1 h until completely dissolved. The above-mentioned cellulose solution was added into the PVA solution, with continuous stirring for 1 h to obtain a uniform solution. Upon cooling to room temperature, 10 mL 9 M KOH solution was added dropwise and continuously stirred for 1 h. After removing bubbles via vacuum, the prepared solution was poured on a polytetrafluoroethylene mold and dried in the fume hood for hours, and a PVA/KOH/MCC polymeric electrolyte membrane was obtained. The prepared PVA/KOH/MCC membrane was rinsed with ethyl alcohol and deionized water several times and placed in a 50% humidity environment for water balance and later use. The membranes loaded with 1, 4, and 7 g MCC are denoted as PVA/KOH/MCC1, PVA/KOH/MCC4, and PVA/KOH/MCC7, in which the weight fractions of MCC are 12, 36, and 50 wt.%, respectively.

### 2.5. Preparation of PVA/KOH/QMCC Alkaline Polyelectrolyte Membrane

Briefly, 7 wt.% NaOH and 12 wt.% urea (urea) were dissolved in deionized water. At room temperature, 0.5, 1, and 2 g MCC were added to NaOH/urea solution separately and were uniformly dispersed by ultrasound and stirred at −12.3 °C until completely dissolved, thus obtaining a colorless, transparent cellulose solution.

Following this, 44 mL epichlorohydrin and 60 mL dodecyl tertiary amine solution were heated and stirred at 60 °C for 2 h to obtain a high-viscosity solution, and the excess epichlorohydrin solution was removed by rotary evaporation. The product was dissolved in acetone solution and stored at 25 °C for 8 h. The filtered product was repeatedly washed with ether and then dried in vacuum at 60 °C for 24 h to obtain the quaternary ammonium salt (QAS).

Subsequently, 1 g quaternary ammonium salt was added into cellulose solution and stirred at 25 °C for 24 h to obtain cellulose quaternary ammonium salt solution. The solution was washed to neutral by ultracentrifugation and purified to obtain quaternary ammonium salt–modified cellulose (QMCC).

Next, 2.15 g PVA was dissolved in 20 mL deionized water, mechanically stirred with 85 °C for 1 h. After which, 1 g glycerol was added into the solution, stirring continuously for 1 h until dissolved completely. The above-mentioned QMCC was added into the PVA solution, stirring continuously for 1 h to obtain a uniform solution. Upon cooling to room temperature, 10 mL 9 M KOH solution was added dropwise and stirred continuously for 1 h. After removing bubbles via vacuum, the resultant solution was placed on a polytetrafluoroethylene mold and dried in a fume hood to obtain the PVA/KOH/QMCC alkaline electrolyte membrane. This membrane was washed repeatedly with ethyl alcohol and deionized water and placed in 50% humidity for water balance and later use. The addition of 0.5, 1, and 2 g MCC in the system are denoted as PVA/KOH/QMCC0.5, PVA/KOH/QMCC1, and PVA/KOH/QMCC2, in which the weight fractions of QMCC are 17, 22, and 30 wt.%, respectively.

### 2.6. Preparation of Zn–Air Electrode

Preparation of air electrode: 0.5 g acetylene black, 0.5 g carbon black, 2.0 g sodium sulfate, 2.0 g KMnO_4_, and 0.5 g vinylidene chloride were ground into a uniform mixture. *N*-methylpyrrolidone solution was added into the mixture dropwise. The mixture was then evenly coated on a nickel foam mesh with a diameter of 12 mm to obtain an air electrode.

Preparation of zinc electrode: 5 g zinc powder and 0.5 g vinylidene chloride were ground into a uniform mixture. *N*-methylpyrrolidone solution was added into the mixture dropwise and stirred continuously to form a mixture with a certain viscosity; the mixture was evenly coated on a nickel foam mesh with a diameter of 12 mm, and the residual *N*-methylpyrrolidone solution was removed by vacuum drying at 100 °C to obtain a zinc electrode.

Copper foil was used as current collector and attached to the surfaces of the zinc and air electrodes. The air electrode, polymer electrolyte membrane, and zinc electrode were listed together in order, and the Zn–air battery was assembled by hot pressing and sealing for later use.

## 3. Results and Discussion

### Preparation and Characterization of Quaternary Ammonium Salt-Modified Cellulose (QMCC)

At room temperature, cellulose was grafted with quaternary ammonium salt with a high positive charge. In this process, cellulose polyhydroxy is replaced by quaternary ammonium group to form quaternized cellulose. Figure 1 shows the reaction mechanism between MCC and quaternary ammonium salt. The chlorine atom on epichlorohydrin was replaced by quaternized amine groups through the reaction of epichlorohydrin and dodecyl tertiary amine. The epoxy ring at the chain end was further grafted on the cellulose via a ring-opening reaction between the epoxy group and hydroxyl group on the cellulose. Finally, quaternary ammonium salt–modified cellulose (QMCC) with higher ion mobility was prepared.

Figure 2 shows the FTIR spectrum of MCC, in which the absorption bands at 3000~3700 and 2850~3000 cm^−1^ are attributed to the stretching vibration of O–H and C–H, respectively. After grafting with QAS, no significant difference was found between QMCC (1 g MCC grafted with 1 g QAS) and MCC, except the band around 3000~3700 cm^−1^. Since the absorption at 3000~3700 cm^−1^ is attributed to the stretch vibration of O–H, the graft of QAS on the OH groups of MCC may change the chemical environment of MCC and lead to a subtle difference in this wave range.

To confirm the grafting of QAS on MCC, SEM analysis with EDX measurement was conducted on the QMCC1-treated ASPE membrane. The EDX spectra shown in Figure 3 revealed the N signal recognized clearly on the cross-section of the ASPE membrane (Figure 3). This N signal provides evidence that the QAS salts were grafted on the MCC and dispersed in the PVA matrix. It was found that the distribution of the N element is quite different in the cross-section of the ASPE membrane. As Figure 3 shows, spots 1’, 2’, and 3’ demonstrate higher N content than spots 1, 2, and 3, indicating the possible aggregation of QMCC1 on spots 1’, 2’, and 3’.

A comparison of the cross-section morphology of the ASPE membranes treated with MCC and QMCC1 was made as Figure 4 shows. Porous morphologies can be clearly recognized in both samples. These irregular pores were formed via the water removal through freeze-drying approach and played an important role in ion transportation. When the ASPE membrane is soaked with water, these pores can act as ion channels to transfer OH^−^ anions. Compared to the surface of PVA/KOH/MCC1, PVA/KOH/QMCC1 shows a rougher surface on which particulate matter appeared. Combined with the result observed in Figure 3, it is speculated that these particulate matters were probably due to the crystallization of quaternary ammonium salt during the freeze-drying process.

In order to evaluate the water-holding capacity of the alkaline polyelectrolyte membrane, a TGA test was carried out. According to Figure 5, it can be clearly found that the weight loss of the alkaline electrolyte membrane is obvious in the temperature range of 45–120 °C, which is attributed to the evaporation of water in the composite membrane. In Figure 5A, it can be found that the water loss of the pure PVA electrolyte membrane is about 60% around 120 °C, while the weight loss of the alkaline polymer electrolyte treated with cellulose is about 70% at 120 °C, and the highest weight loss is about 75% when the cellulose loading is 4 g. This is mainly because the introduction of microcrystalline cellulose leads to a large number of hydroxyl groups in the matrix. These hydroxyl groups can form a large number of hydrogen bonds with water and provide favorable conditions for the transfer of OH^−^ in the matrix and, thus, improve ionic conductivity (see Figure 6). When the cellulose loading reaches 7 g, the water retention capacity of the polymer electrolyte decreases. This result is probably caused by the further increased microcrystalline cellulose content. The excessive MCC will agglomerate and reduce the specific surface area, impairing the contact between cellulose and water molecules, further reducing the water retention capacity. In this context, a TGA test of PVA/KOH/QMCC was carried out. Figure 5B shows that an improved water retention capacity for PVA/KOH/QMCC system, in which the weight loss reached up to 80%, was observed when the QMCC loading reached 2 g. The introduction of QAS on cellulose significantly enhanced the water retention capacity and was beneficial for the improvement of the ionic conductivity of the ASPE membrane.

The AC impedance curve of the composite membrane was tested using an electrochemical workstation, and the test results are shown in Figure 6A,C. From the AC impedance curve, the volume resistance, R_b_ (the intersection of the curve and the horizontal axis), of the polyelectrolyte membrane is obtained, and the ionic conductivity of the polyelectrolyte is calculated by Equation (1).
σ = L/(R_b_ × A)(1)
in which σ is the ionic conductivity (mS cm^−1^), L is the membrane thickness (cm), R_b_ is the volume resistance (ω), and A is the membrane surface area (cm^2^).

Figure 6B shows the conductivity curves of the PVA/KOH and PVA/KOH/MCC composite polyelectrolyte membranes. It can be seen that the conductivity of the composite membrane increases at first and then decreases. When cellulose is not added, the ionic conductivity of the PVA/KOH electrolyte is 50 mS cm^−1^, while when cellulose loading reaches 4 g, the ionic conductivity of the composite membrane reaches the highest value of 91 mS cm^−1^. This is mainly because MCC increases the alkali resistance and water retention ability of the PVA matrix. With the increase of MCC content, the decreased PVA ratio impaired the transportation of OH^−^, leading to a decrease in conductivity. Compared to the MCC, the QMCC endows the ASPE membrane with a remarkable improvement in conductivity (Figure 6D). The highest value of 238 mS cm^−1^ was obtained when QMCC loading reached 1 g, which is almost five times of the conductivity value of the PVA/KOH membrane. Clearly, the intrinsic hydroxyl groups on the QMCC favor water absorption; moreover, the quaternary ammonium salts grafted on the MCC chains play an important role in absorbing higher contents of OH^−^ via electrostatic interaction. Similar to the results obtained in Figure 6B, more loading of QMCC would not benefit the conductivity, and a declined value was obtained when the QMCC reached 2 g.

In order to evaluate the electrochemical stability window, the cyclic voltammetry curves of polyelectrolyte membranes were measured using an electrochemical workstation. As Figure 7A shows, the PVA/KOH composite polyelectrolyte membrane demonstrates an electrochemical stability window of about 1.0 V, and there is no significant change after the introduction of MCC. These curves have good symmetry and no redox peak, showing good electrochemical stability and potential in recycling at high current density. Compared to the PVA/KOH/MCC alkaline electrolyte membranes, the quaternary ammonium salt–modified membranes demonstrate a different electrochemical behavior. It can be found in Figure 7B that there is a larger area surrounded by the cyclic voltammetry curve of the PVA/KOH/QMCC membrane, indicating a higher specific capacity for the quaternary ammonium salt–modified ASPE compared to the neat one. Moreover, the stable voltage of the alkaline electrolyte membrane modified by quaternary ammonium salt is −0.5 to +0.5 V, showing no significant difference compared to that of the neat one. This electrochemical stability window shown in Figure 7B suggests that the membrane made with PVA/KOH/QMCC can meet the application requirements of a Zn–air battery.

A Zn–air battery was then assembled using the PVA/KOH/QMCC1 membrane as the alkaline solid polyelectrolyte. Figure 8A shows the mechanical properties of the PVA/KOH/QMCC1 membrane. Compared to the PVA/KOH membrane, it can be found in Figure 8A that the tensile stress was increased by almost two times after the addition of QMCC1, combined with an increased breaking strain from 75% to 100%. The improved mechanical properties were attributed to the intrinsic high modulus of the microcrystalline cellulose (MCC) and the interactions between PVA and MCC via the hydrogen bonds. Clearly, the increased breaking strain and tensile stress are beneficial for the battery design. Figure 8B,C show the discharge polarization curves and the power density curves of the Zn–air battery using PVA/KOH and PVA/KOH/QMCC1 as the ASPE membrane, respectively. The current density and power density of the zinc–air battery with PVA/KOH/QMCC1 are obviously higher than those of zinc air battery with PVA/KOH. When the open circuit voltage is 0.9 V, the corresponding current density can reach 128 mA cm^−2^, and the peak power density can reach 48 mW cm^−2^. Figure 8D shows the discharge curves of zinc–air batteries using PVA/KOH and PVA/KOH/QMCC1 as a polyelectrolyte, under a condition of current density of 20 mA cm^−2^. The rapid voltage drop within 0–2 min during the discharge process is due to self-discharge, and then the rapid voltage drop reaches a stable state. In this progress, the zinc electrode is consumed constantly, leading to a rapid voltage drop. Compared with the initial voltage of 0.85 V and discharge time of 56 min for the zinc–air battery using PVA/KOH as the ASPE membrane, the zinc–air battery using PVA/KOH/QMCC1 as the ASPE membrane demonstrated an initial voltage of 0.99 V and a discharge time of 73 min, indicating a longer discharge time and an improved performance. Figure 8E–G shows the assembled flexible zinc–air battery and its application in light-emitting diode (LED, 1.8 V). As Figure 8F,G shows, after being bent 90 degrees, the battery using PVA/KOH/QMCC1 as the ASPE membrane still makes the LED bulb glow.

## 4. Conclusions

In this paper, quaternary ammonium salt–grafted cellulose was prepared by dissolving microcrystalline cellulose with a green solvent NaOH/urea solution and reacted with quaternary ammonium salt (QAS). A film was then prepared via mixing QAS-grafted cellulose with PVA in KOH solution. Taking advantage of the hydrophilicity and alkali resistance of the cellulose, the improved water retention, ionic conductivity, and alkali resistance properties were obtained for the MCC added PVA/KOH membrane. Further increased ionic conductivity and specific capacity can be achieved via the modification of the cellulose with quaternary ammonium salt. Experimental results show that after the addition of QMCC, a remarkable increase in ionic conductivity from 50 to 238 mS cm^−1^ was achieved for the PVA/KOH electrolyte system. As reinforcing fillers, celluloses can also improve the mechanical properties of polyelectrolyte membranes, whereby a tensile stress increased by two times was obtained. Moreover, the PVA/KOH/QMCC polyelectrolyte membrane studied here demonstrates a feasibility in assembling a flexible Zn–air battery and shows promising potential in the development of flexible electronic devices.

## Figures and Tables

**Figure 1 polymers-13-00009-f001:**
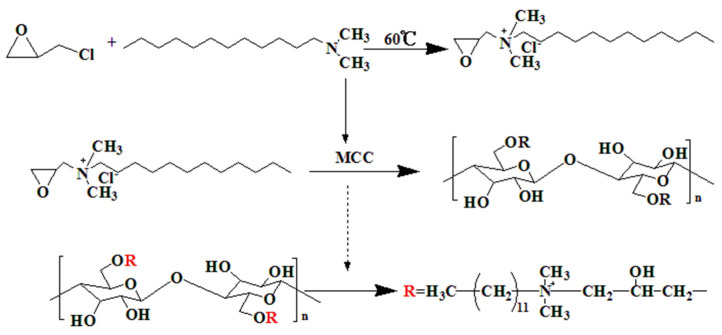
Preparation of quaternary ammonium salt–modified cellulose (QMCC) via grafting quaternary ammonium salt (QAS) on cellulose.

**Figure 2 polymers-13-00009-f002:**
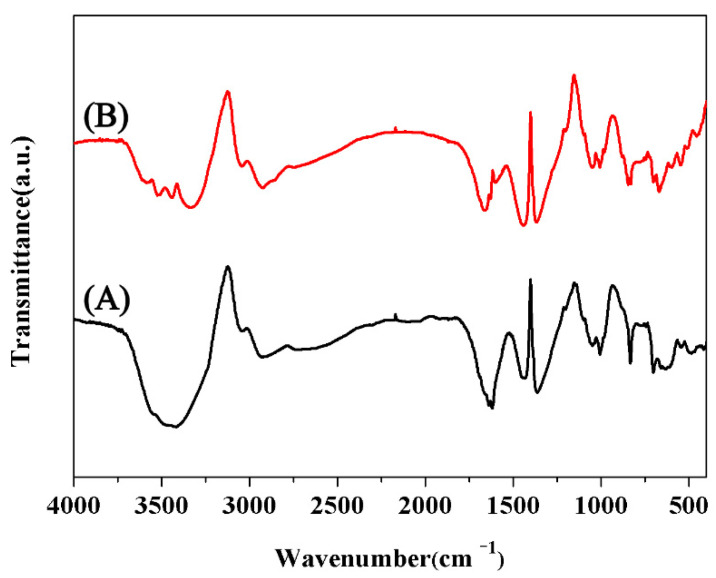
FTIR curves of (**A**) microcrystalline cellulose (MCC) and (**B**) QMCC1.

**Figure 3 polymers-13-00009-f003:**
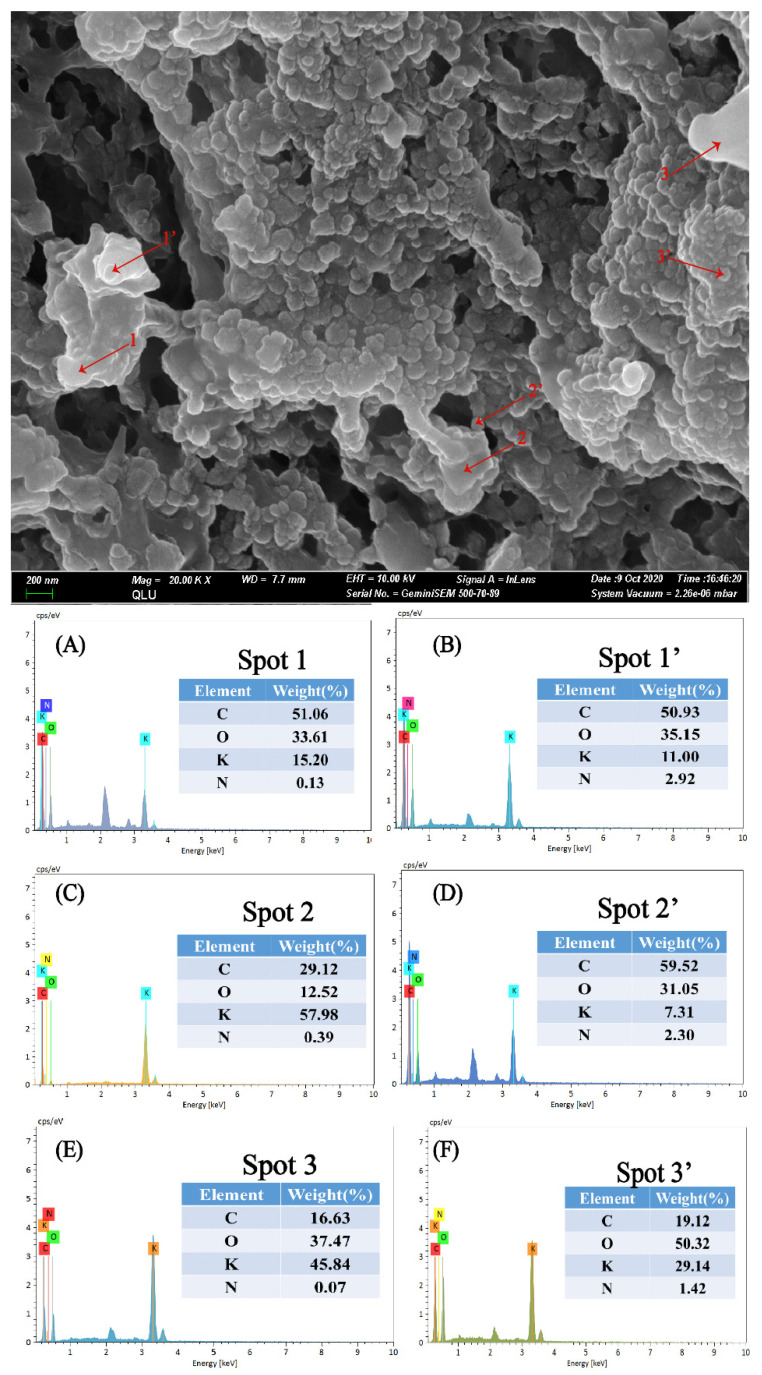
(**A**) The EDX curve of PVA/KOH/QMCC alkaline polymer electrolyte membranes at Spot 1. (**B**) The EDX curve of PVA/KOH/QMCC alkaline polymer electrolyte membranes at Spot 1’. (**C**) The EDX curve of PVA/KOH/QMCC alkaline polymer electrolyte membranes at Spot 2. (**D**) The EDX curve of PVA/KOH/QMCC alkaline polymer electrolyte membranes at Spot 2’. (**E**) The EDX curve of PVA/KOH/QMCC alkaline polymer electrolyte membranes at Spot 3. (**F**) The EDX curve of PVA/KOH/QMCC alkaline polymer electrolyte membranes at Spot 3’.

**Figure 4 polymers-13-00009-f004:**
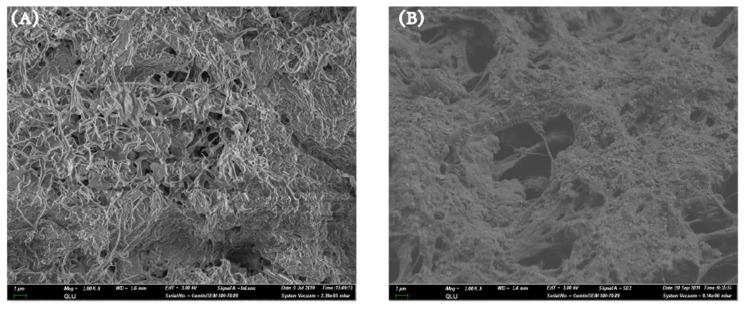
The SEM microphotographs of cross-section for alkaline polymer electrolyte membranes of (**A**) PVA/KOH/MCC1 and (**B**) PVA/KOH/QMCC1.

**Figure 5 polymers-13-00009-f005:**
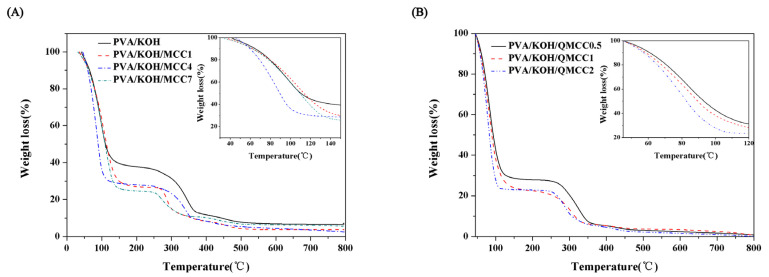
(**A**) TGA curves of PVA/KOH alkaline electrolyte membranes with various contents of MCC. (**B**) TGA curves of PVA/KOH/QMCC alkaline electrolyte membranes.

**Figure 6 polymers-13-00009-f006:**
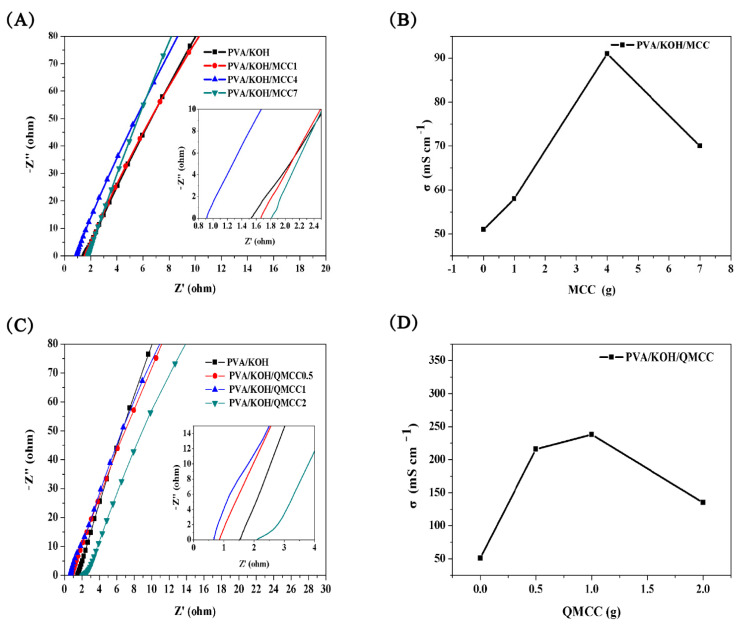
(**A**) AC impedance curve of PVA/KOH and PVA/KOH/MCC alkaline electrolyte membranes. (**B**) Conductivity of PVA/KOH/MCC alkaline electrolyte membranes. (**C**) AC impedance curve of PVA/KOH/QMCC alkaline electrolyte membranes. (**D**) Conductivity of PVA/KOH/QMCC alkaline electrolyte membranes.

**Figure 7 polymers-13-00009-f007:**
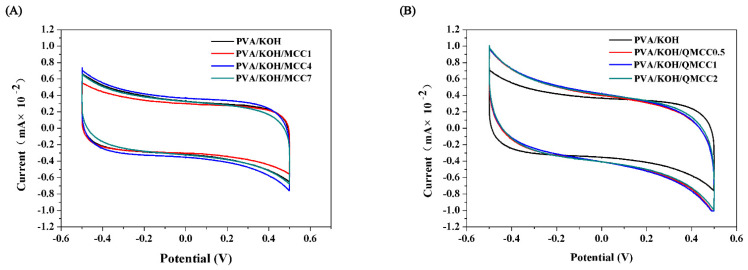
(**A**) Cyclic voltammetry curves of PVA/KOH/MCC alkaline electrolyte membranes. (**B**) Cyclic voltammetry curves of PVA/KOH/QMCC alkaline electrolyte membranes.

**Figure 8 polymers-13-00009-f008:**
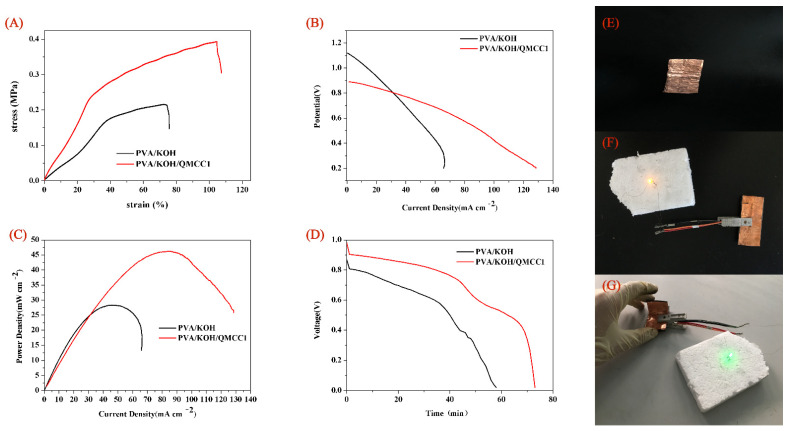
(**A**) The stress–strain curves of PVA/KOH and PVA/KOH/QMCC1. (**B**) Polarization curves of Zn–air battery using PVA/KOH and PVA/KOH/QMCC1 as the alkaline solid polyelectrolyte (ASPE) membrane. (**C**) Power density curves of Zn–air battery using PVA/KOH and PVA/KOH/QMCC1 as the ASPE membrane. (**D**) Discharge curve of Zn–air battery with a current density of 20 mA cm^−2^ using PVA/KOH and PVA/KOH/QMCC1 as the ASPE membrane. (**E**) Assembled Zn–air battery. (**F**) and (**G**) Application of Zn–air battery in LED bulb.

## Data Availability

Please refer to suggested Data Availability Statements in section “MDPI Research Data Policies” at https://www.mdpi.com/ethics.

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
