# Peer review of "Preparation of Alkaline Polyelectrolyte Membrane Based on Quaternary Ammonium Salt–Modified Cellulose and Its Application in Zn–Air Flexible Battery"

_polymers, 2020, doi:10.3390/polym13010009_

Round 1

Reviewer 1 Report

The manuscript "Preparation of alkaline polyelectrolyte membrane  based on quaternary ammonium salt modified cellulose and its application in Zn-Air flexible battery"  describes a new method of preparation of alkaline polyelectrolyte membrane. The research may be of interest for the readers of the "Polymers" journal, however, some points need to be clarified.

1) Lines 176-188 Please, specify exact amounts of the components, used for preparation of electrodes slurry. 

2) Please, provide the manufacturer and model of the LED used for Zn-air battery test.

3) Provide some precise data on Zn-air battery (EMF, power/current and volt/ampere characteristics)

Minor points:

Line 43 - replace "polymer solid electrolyte (SPE)" with "solid polymer electrolyte (SPE)"

Line 113 - please, correct the typos in the frequency values - superscript is missing.

Line 121 -  at 10 mV s-1 superscript is missing

Reviewer 2 Report

The manuscript compares polymer electrolytes with different mechanical supports, MCC, and QMCC. Although their results show the improvements in mechanical properties, condcutivities. I can't see any insights from the manuscript and the discussions are too general. I think the manuscript is not suitable for publication. My other comments:

  1. The FTIR is not convincing as evidence for the success of grafting, the curves are two close, do they have other evidence? 
  2. Why the authors label the sample with loading (e.g.4g), not weight fraction?
  3. The comparison of the Zn-air battery should be quantified, not by simple demonstration, therefore the electrochemical performance tests performances are necessary. 
